# Developing Problematic Performance Value Scores: Binding Routine Activity Performance, Environmental Barriers, and Health Conditions

**DOI:** 10.3390/ijerph21060764

**Published:** 2024-06-13

**Authors:** Jimin Choi, JiYoung Park

**Affiliations:** 1Center for Inclusive Design & Environmental Access, School of Architecture & Planning, University at Buffalo, SUNY, Buffalo, NY 14214, USA; jchoi32@buffalo.edu; 2Department of Urban and Regional Planning, University at Buffalo, SUNY, Buffalo, NY 14214, USA; 3MUREPA Korea, Seoul 13640, Republic of Korea

**Keywords:** problematic performance value, environmental barrier, health conditions, universal design, multinomial logit model

## Abstract

Background: Community design features, such as sidewalks and street crossings, present significant challenges for individuals with disabilities, hindering their physical performance and social integration. However, limited research has been conducted on the application of Universal Design (UD) to address these challenges, particularly concerning specific demographic groups and population cohorts. Understanding the influence of environmental features on physical performance is crucial for developing inclusive solutions like UD, which can enhance usability and social integration across diverse populations. Objective: This study aims to bridge this gap by investigating the complex relationships between environmental barriers, health conditions, and routine activity performance. An index was developed to evaluate users’ UD performance based on functional capacity, providing scientifically rigorous and objectively measured evidence of UD effectiveness in creating inclusive built environments. Method: Using data from the Problematic Activities Survey (PAS) conducted in the U.S., Canada, and Australia and targeting individuals with and without functional limitations, multinomial logit models were employed to estimate the probabilities of encountering performance problems. This analysis led to the development of the Problematic Performance Value (PPV) score. Results: The results demonstrated significant disparities in PPVs across various health conditions, particularly concerning curb ramps. Individuals facing mobility issues in their legs/feet, arms/hands, or back/neck encounter more pronounced challenges, especially when curb ramps lack proper design elements. Similarly, individuals with vision impairments face heightened difficulties with traffic signals, particularly due to issues with audible signal systems. These findings underscore the importance of addressing micro-level environmental challenges to accommodate individuals with varying functional capacities effectively. Conclusions: By providing insights into the most problematic daily activities encountered by diverse populations, the PPV score serves as a valuable indicator for guiding environmental design improvements and promoting equitable space usage. This can be used to guide improved UD solutions and decide areas of concentration by providing generalized information on specific environmental features that contribute to user performance.

## 1. Introduction

The role of community design in determining health outcomes has been examined in a growing number of studies over the past decades [1,2,3,4,5]. However, the intricate relationships between environment-related features and physical activity and health remain complex [5,6,7,8], further complicated by personal factors such as age and functional capacity, posing challenges for policy interventions [9,10]. Given the absence of standard methods, the development of a framework to clarify the complex associations between environmental barriers and health conditions is critical, which supports decision makers in effectively allocating resources.

Among the community design features, infrastructure such as sidewalks and street crossings presents significant challenges for millions of Americans, particularly those with disabilities [11,12]. Despite the importance of accessing homes and community support, nearly half of Americans aged 50 and older struggle to cross the main roads near their residences [13]. The lack of “first/last mile” connectivity—poor pedestrian access to and from fixed route stops—poses a significant obstacle along the travel chain [14,15]. As a result, approximately 3.6 million individuals with disabilities are unable to leave their homes due to transportation difficulties [16]. Therefore, there is a need to understand how these environmental features in the first/last mile pathways exacerbate or mitigate the gap between an individual’s physical performance and the physical demands of a given activity [17,18].

Universal Design (UD) offers inclusive solutions that benefit all populations by addressing the current deficiencies in the built environment. UD empowers individuals by enhancing usability and safety beyond minimal code compliance and supporting a broader range of disabilities, including those with severe impairments [19,20]. Effective implementation of UD can improve function and social integration through the design of physical, virtual, and social environments [21]. Therefore, this research aims to implement UD concepts in developing the framework. Despite the growing body of research on UD’s efficacy, limited studies exist due to the significant time and effort required to measure user performance for specific design criteria and the lack of consensus on its application and evaluation [22,23,24]. Thus, developing a method to evaluate the user performance of UD is essential for creating more measurable, evidence-based guidelines for implementing UD concepts. 

UD has progressed from its initial emphasis on usability to encompass broader dimensions, such as health, wellness, and social participation [25]. The concept of “usability” evaluates UD effectiveness by examining specific activities and an individual’s interaction with the environment [26,27,28]. In rehabilitation and disability studies, usability integrates an individual’s assessment of the extent to which desired activities can be performed within a given environment, comprising three components: personal functional capacities, environmental barriers, and performed activities [29]. Incorporating usability concepts, Figure 1 illustrates measuring user performance on specific environmental barriers relative to personal capacity. 

Environmental barriers significantly hinder the ability of individuals with functional limitations to engage in activities effectively, particularly when these barriers exacerbate existing challenges. Curb ramps and inaccessible sidewalks exemplify such built environment barriers. Individuals with mobility disabilities may struggle to navigate uneven sidewalks and steep slopes, requiring excessive physical exertion [30,31,32,33]. Uneven pavement, puddles, and poor drainage further complicate mobility for individuals with visual and motor impairments [34]. These barriers can substantially reduce the participation of individuals with disabilities in activities outside the home [35,36,37]. Conversely, environments designed to be accessible and supportive can mitigate these challenges, enhancing usability. Despite many studies identifying environmental barriers that limit mobility and physical activity among individuals with disabilities, limited sample sizes have hindered the comprehensive quantification of these relationships. There is a consensus that further research is needed to better understand the interaction effects between disability type and built environment features [38]. However, such research can be time-consuming and resource-intensive, which limits the current body of knowledge on how different types of functional limitations are impacted by built environmental barriers and facilitators. 

Understanding the impact of environmental barriers and health conditions on specific activities will provide insights that support evidence-based practices in UD [39], enhancing usability for all users regardless of differences in functional capacity, ability, or circumstances. This study, therefore, contributes to addressing usability problems individuals encounter when engaging in various activities amid different environmental barriers, with a particular focus on public street settings. By developing an index to evaluate user performance based on functional capacity, the study primarily explores the interplay among three usability components to understand their collective impact on user performance. 

A number of environmental assessment tools exist, primarily focusing on measuring the environmental performance of buildings while indirectly assessing usability aspects. For example, tools such as the Comprehensive Assessment System for Building Environment Efficiency (CASBEE) and the Sustainable Building Tool (SBTool 2020) provide comprehensive frameworks for evaluating environmental performance and sustainability, addressing some functionality and usability aspects [40]. However, these tools do not fully capture the scope of user interaction and experience with the built environment.

Attempts to evaluate the effectiveness of UD implementation have been limited. The innovative solutions for Universal Design (isUD) program offers a systematic evaluation of UD applications in buildings, furnishing project teams with a detailed checklist to promote ongoing inclusivity [41]. However, further research is necessary to substantiate the business case for UD and to evaluate the effectiveness of isUD as a design and environmental audit tool. Additional post-occupancy evaluations are essential to determine building performance and user satisfaction. In addition, subjective evaluation tools may be required alongside objective checklists to gather insight from individuals with disabilities [42]. 

The Problematic Activities Index (PAI) evaluated the effectiveness of UD by measuring individuals’ ability to perform daily activities and their performance levels. A survey within PAI identified environmental issues hindering activities, prompting participants to report the frequency of encountering difficulties with the proposed design [39]. Nonetheless, the PAI approach merely elucidated collected data without generalizing PAI scores or their applicability in predicting problematic activity scores across different contexts. 

Utilizing the dataset from the PAI assessments conducted in 2008, this study could successfully develop a tool for measuring users’ UD performance based on functional capacity. Unlike previous tools that primarily focus on environmental performance and sustainability, this study provided a more comprehensive and scientifically rigorous approach to assessing UD effectiveness. It employed robust quantitative methodologies to generalize findings across different contexts, offering objective evidence that can guide the development of more inclusive and user-centered urban design policies.

## 2. Methods

### 2.1. Data Collection

An anonymized and non-personally identifiable dataset was acquired from the Problematic Activities Survey (PAS) conducted in 2008. With the IRB approval, this survey examined the influence of the built environment on individuals’ routine activities. The original PAS data aimed to understand how the built environment affects adults’ ability to perform routine activities and targeted five user populations: individuals with mobility, hearing, vision, or high-functioning cognitive limitations, as well as those without any functional limitations. The PAS data were collected through internet bulletins, newsletters, and listservs of national organizations serving people with functional limitations, independent living centers, and professional rehabilitation organizations. The primary focus of the survey was the U.S.; however, the survey was also accessible to individuals residing outside the U.S. who were part of the organizational network. After obtaining informed consent, the survey gathered information on (1) the frequency with which specific activities were problematic (i.e., always, sometimes, and never) across three environmental settings (i.e., public streets, public buildings, and residential environments), and (2) basic demographic characteristics, including age, gender, income, type of residence, and physical or cognitive conditions affecting their routine activities. The PAS questionnaires covered 11 activities for the public streets setting (e.g., using curb ramps, detecting public sidewalks, using transit stops, etc.), 23 activities for the public building setting (e.g., using parking areas, using paths of travel to entrances, using signs or maps, using hallways, etc.), and 20 activities for residential environments (e.g., using entrances, hallways, countertops, etc.). For each problematic activity, participants were asked to specify the reasons for the encountered issues. Health conditions were measured in seven categories (i.e., mobility of arms/hands, legs/feet, and back/neck, hearing, vision, cognition, and other) and rated based on how often the condition affected their ability to perform routine activities (i.e., always, sometimes, and never).

### 2.2. Environmental Barriers

This study analyzed data collected from the PAS to evaluate the effectiveness of environmental design for various activities, both within and across different participant health conditions. The focus was specifically on intersection activities on public streets, including using curb ramps (Curb), using traffic signals (Signal), and using crosswalks (Crosswalk) to identify street safety issues related to individual health conditions. The rationale behind selecting intersection activities on public streets was their prevalence in individuals’ daily routines, ensuring a broad spectrum of environmental features and capturing wide variations among these features.

Open-ended responses about reasons for problematic performances were examined to identify frequently mentioned environmental barriers. Researchers used open and focused coding to identify common patterns in the qualitative data [43,44]. Five themes emerged as environmental barriers for the Curb activity: (a) missing curb ramps, (b) curb design issues, such as slope, size, and groove at the bottom, (c) curb location and position, (d) curb surface, and (e) other issues, such as tactile marking problems and obstacles blocking access. For the Signal activity, four environmental themes were identified: (a) improper signal duration, (b) issues with signal buttons, (c) lack of audible signals, and (d) other issues, such as difficulty locating traffic signals, signal conflicts, and non-uniform systems. Lastly, three themes were identified for the Crosswalk activity: (a) problems with marking and design, (b) uneven surfaces, and (c) other issues, such as location, length, and obstacles.

Participants who reported never having problems with their performance were not required to explain why, and their responses were coded as having “no barrier”. Similarly, participants who identified problems but did not specify environmental barriers were also coded as having “no barrier”.

### 2.3. Participant Demographic Characteristics

In total, 488 samples were collected from the PAS in the public street setting. Among the respondents, approximately 33% (*n* = 161) were male. The age distribution was as follows: 23% (*n* = 112) were younger than 29, 12.5% (*n* = 61) were aged 30 to 39, 19.7% (*n* = 96) were aged 40 to 49, 24.6% (*n* = 120) were aged 50 to 59, 14.5% (*n* = 71) were in their 60s, and 5.1% (*n* = 25) were older than 70. Three respondents did not report their age. Income levels were evenly distributed across low (less than USD 24,999; 25.6%; *n* = 125), middle (USD 25,000–USD 74,999; 33.8%; *n* = 165), and high (USD 75,000 or more; 27.4%; *n* = 134) annual household incomes, with 13.1% (*n* = 64) not reporting their income. Regarding housing type, 57% (*n* = 280) lived in a single-family home, while the remainder lived in either a two-family house, apartment, condo, townhouse, or mobile home. Only 1.4% did not answer this question. Most survey respondents (83.3%) resided in the U.S., with 4.3% in Canada and 10.9% in Australia, while 1.4% did not answer the zip code question. In terms of problematic health conditions, a high percentage of respondents frequently had issues with mobility. Specifically, 32% “always” and 19.9% “sometimes” experienced problems with leg/foot mobility. Additionally, 16.8% “always” and 27.0% “sometimes” had problems with arm/hand mobility, while 13.5% “always” and 33.1% “sometimes” had back/neck mobility issues. Respondents who reported “always” having hearing, vision, and other health condition problems ranged from 11% to 15%. Only 4.8% of respondents reported “always” having mental and/or cognitive issues.

### 2.4. Multinomial Logit Model and the Problematic Performance Value Score

Based on the original PAS data, each problematic activity was analyzed using multinomial logit (MNL) models. A new usability index, termed the Problematic Performance Value (PPV) score, was developed. The MNL model is widely used for multi-discrete dependent variables, retaining the probability of multi-chotomous selection within a range of 0 and 1. The MNL regression form is presented below [45].
(1)$$\log\left(\frac{\mathrm{Prob}(y=j)}{\mathrm{Prob}(y=J)}\right)=\sum_{k=1}^{K}\beta_{jk}x_{k}$$

Hence, probabilities from the MNL regression can be obtained via
(2)$$\mathrm{Prob}(y=J)=\frac{1}{1+\sum_{j=1}^{J-1}e^{\sum_{k=1}^{K}\beta_{jk}x_{k}}}$$
(3)$$\mathrm{Prob}\left(y=j\right)=\frac{e^{\sum_{k=1}^{K}\beta_{jk}x_{k}}}{1+\sum_{j=1}^{J-1}e^{\sum_{k=1}^{K}\beta_{jk}x_{k}}}$$

As described in Equation (1), the MNL approach assumed the choice of one of three problematic performance responses: “always”, “sometimes”, and “never”. In the MNL applications, several independent variables were transformed from their original data codes to new codes. This data reformation was necessary because all variables were in a single-answer multiple-choice format, requiring continuous or modified dummy variables for the MNL regressions.

The selected variables are summarized in Table 1. Each PAS activity model used median age, median age squared, income categories (low, middle, and high income and unknown/prefer not to answer), housing type, and one health condition as independent variables. For each activity, the chosen health condition was combined with other personal variables to measure problematic performance. A total of seven MNL models were run per PAS activity. For example, in constructing the PPV model for the Curb activity, models M-CURB 1 to 7 included the following health conditions, respectively: (1) arms/hands mobility, (2) legs/feet mobility, (3) back/neck mobility, (4) hearing, (5) vision, (6) mental/cognitive, and (7) other. Therefore, a total of 21 MNL models (3 activities X 7 health conditions) were ultimately run.

While each model was confined to a specific environmental barrier, detailed environment-related design features were not included in developing the PPV score due to limitations in obtaining structured data from the PAS. Instead, the relationship between PPV and environmental barriers is discussed in a later section.

Using the individual-based PPV scores metric for each PAS activity derived from the MNL results, probabilities of experiencing problems in activity performance (i.e., always, sometimes, and never) were calculated. The PPV was created using the individual’s selection of one health condition along with other demographic variables (i.e., P_h_(Always), P_h_(Sometimes), and P_h_(Never)). The individual probability of problematic performance was decomposed to the structures of probability constrained by each health condition level, recalculating each probability of P_ij_. For example, the probabilities of problematic performance being “always”, “sometimes”, and “never” were recalculated by applying one MNL model constrained to a health condition being “always” (as highlighted in Table 2). Additional MNLs, constrained by each health condition level, provided sub-probabilities for the individual probability of problematic performance.

The sub-probabilities for each health condition level (e.g., P_11_, P_12,_ and P_13_) were summed to one. Therefore, by multiplying the P_h_(Always) with the estimated sub-probabilities, consistent with the PAI index calculation, the individual PPV score was calculated as follows.
PPV = (P_11_ + 0.5 × P_21_ + 0 × P_31_) × 1 × P_h_(Always) + (1 × P_12_ + 0.5 × P_22_ + 0 × P_32_) × 0.5 × P_h_(Sometimes) + (1 × P_13_ + 0.5 × P_23_ + 0 × P_33_) × 0 × P_h_(Never) = (P11 + 0.5 × P_21_) × 1 × P_h_(Always) + (1 × P_12_ + 0.5 × P_22_) × 0.5 × P_h_(Sometimes)(4)
where P_ij_ represents the sub-probability constrained to the health condition level.

The following section summarizes the MNL analysis, focusing on the mean and median PPV scores, and compares the seven health conditions and environmental barriers within the selected Curb, Signal, and Crosswalk activities.

## 3. Results

### 3.1. MNL Results

Models applied to the activities of *Curb* (M-CURB), *Signal* (M-SIGNAL), and *Crosswalk* (M-CWALK) explored the effects of age, sex, income, housing type, and health conditions on varying performance levels. Each model, ranging from 1 to 7, included different health conditions (i.e., arms/hands mobility, legs/feet mobility, back/neck mobility, hearing, vision, mental/cognitive, and other) categorized into levels of “always” (HC_al) and “sometimes” (HC_st). For the M-CURB model, age was a statistically significant factor that increased the frequency of problematic performance (see Table 3). Additionally, being in the high-income group and consistently having problems with various health conditions, both physical and cognitive, were found to significantly impact curb ramp activity performance.

As presented in Table 4, the M-SIGNAL results showed that the median age variable across different health condition models significantly impacted the likelihood of “sometimes” experiencing problems compared to “never” experiencing problems when using traffic signals. Specifically, Model 5 (Vision) of M-SIGNAL demonstrated that individuals who “always” had vision problems were more likely to “always” or “sometimes” encounter difficulties using traffic signals.

Similarly, as shown in Table 5, the median age variable across different health condition models had a significant relationship with “sometimes” experiencing problems using crosswalks in the M-CWALK model. The high-income group was also found to significantly affect the likelihood of “always” or “sometimes” experiencing problems compared to “never” having problems using crosswalks. Models 1, 2, 3, and 5 in Table 5 revealed that individuals with mobility issues in their arms/hands, legs/feet, back/neck, or those with vision impairments, respectively, were more likely to “always” or “sometimes” encounter problems when using crosswalks.

### 3.2. Problematic Performance Value Scores by Activities and Health Conditions

Possible PPV ranges from 0 to 1. PPV scores for each health condition across the selected activities ranged from 0.0007 to 0.4601. Mean and median PPVs for various health conditions are presented in Table 6. In both Curb and Crosswalk activities, the mean/median PPVs for arms/hands mobility, legs/feet mobility, back/neck mobility, and vision surpassed those for hearing, mental, and other conditions. Conversely, in the Signal activity, mean/median PPVs for vision exhibited the highest values. Moreover, the mean and median PPVs for hearing generally remained lower compared to the PPVs for other health conditions across the diverse activities.

Variations in the mean/median PPVs were also evident across different activities within the same health condition. For example, the mean/median PPVs for arms/hands mobility, legs/feet mobility, and back/neck mobility in the Curb activity were about 0.03 higher than those in the Signal or Crosswalk activities. These findings suggest that individuals experiencing mobility issues with their arms/hands, legs/feet, and back/neck may encounter more pronounced challenges in the Curb activity. Conversely, the mean PPV for hearing remained relatively consistent across different activities. Similarly, the mean PPV for vision, mental/cognitive, or other conditions exhibited consistency within the Curb and Signal activities but registered lower values for the Crosswalk activity.

### 3.3. Problematic Performance Value Scores by Environmental Barriers and Health Conditions

The relationships between various environmental barriers and health conditions for specific activities using median/mean PPVs were analyzed. Across all curb ramp-related environmental barriers, the mean/median PPVs were highest for legs/feet mobility, followed by arms/hands mobility and back/neck mobility. This suggests that respondents facing challenges with mobility in their legs/feet, arms/hands, and back/neck are more likely to encounter significant difficulties when dealing with issues such as missing curb ramps, improper curb size or slope, or uneven curb surfaces compared to those with other health conditions. For example, one participant commented, “curb ramps are sometimes too steep for my power wheelchair (or non-existent!).”

In the case of Curb activities on public streets, as shown in the top figure of Figure 2, the mean/median PPVs for arms/hands mobility, legs/feet mobility, and back/neck mobility were higher when curb ramps were absent or had design or surface issues. Conversely, issues related to curb location or position appeared to be less problematic for individuals with challenges in arms/hands mobility, legs/feet mobility, and back/neck mobility compared to other curb-related environmental barriers. PPVs for hearing, vision, mental/cognitive, and other health conditions showed similar patterns across different environmental barriers.

The mean/median PPV scores for vision in the Signal activity, as shown in Figure 2, were consistently higher across various environmental barriers compared to PPVs for other health conditions. This difference was particularly pronounced with the audible signal system, indicating that individuals with vision impairment may face increased challenges in situations where there is no audible signal system or when the audible signal is malfunctioning. For example, one respondent noted, “*Although I’ve learned to use hearing most of the time, I am not always able to determine when a signal would indicate safe crossing. If a crossing has an audible signal, I find it much easier.*”.

Furthermore, it is important to note that the median PPVs for arms/hands, legs/feet, back/neck, mental/cognitive, and other conditions related to signal buttons were generally higher than for other traffic signal-related barriers. This suggests that individuals may encounter significant difficulties when using the traffic signal button (e.g., due to the signal button being unreachable or hard to push), irrespective of their health conditions.

In the case of crosswalk markings, illustrated in the bottom figure of Figure 2, individuals with vision impairment exhibited higher median PPVs, indicating they may encounter greater challenges when encountering situations where the crosswalk is missing, poorly marked, or difficult to detect. Similarly, individuals with legs/feet mobility or back/neck mobility issues showed higher mean/median PPVs for crosswalk surface issues, suggesting increased difficulty when navigating crosswalks with uneven surfaces.

## 4. Discussion

This study investigated the impact of specific environmental barriers and health conditions on routine activity performance, a relationship that has not been quantitatively identified over the past decades [5,6,7,8] due to various personal factors [9,10]. Given the absence of standard methods, developing a framework to clarify the complex associations between environmental barriers and health conditions is critical.

The results of this study indicate that various demographic and health condition factors significantly impact the performance levels in curb, signal, and crosswalk activities. Age emerged as a statistically significant factor across all models, with older individuals experiencing higher frequencies of problematic performance. This finding aligns with previous studies, which demonstrate that the natural decline in physical and cognitive abilities with age makes it more challenging to navigate urban environments that are not universally designed. Reducing environmental burdens may increase independence in daily activities and overall well-being [46,47]. Additionally, physical and cognitive health conditions consistently presented significant challenges across all activities. Mobility impairments in the arms/hands, legs/feet, back/neck, and vision issues were particularly impactful, underscoring the need for inclusive design that addresses these specific challenges.

The developed PPV score served as a usability index, quantifying the challenges individuals faced when engaging in activities with particular health limitations. A higher PPV indicated greater challenges in performing activities within a defined environment. Comparing PPVs across different activities revealed that individuals encountered more severe difficulties using curb ramps regardless of health conditions, emphasizing the need to prioritize curb ramps to reduce environmental barriers. Targeted improvements in these areas may significantly enhance usability. By investigating different activities, the PPV can serve as a valuable indicator for identifying the most problematic daily activities encountered by diverse populations.

Environmental barriers further exacerbated the difficulties individuals encountered. For example, individuals with mobility issues in their legs/feet, arms/hands, or back/neck faced more pronounced challenges with the absence or improper design of curb ramps compared to those with different health conditions. Similarly, individuals with vision impairments faced heightened difficulties with inadequate audible signal systems and poorly marked crosswalks. These findings underscore the importance of addressing micro-level environmental challenges to ensure effective accommodation for individuals with varying functional capacities. This highlights the need for collaborative efforts among planners, designers, and practitioners in fostering inclusive environments.

The PPV tool can significantly enhance the usability and practicality of UD in various environmental settings. It helps identify specific environmental barriers that hinder routine activities among individuals with different functional limitations. By assigning a score to the frequency and severity of problematic activities, the PPV tool quantifies the extent of difficulty experienced by users. This quantification aids in understanding which barriers are most problematic and require immediate attention. In addition, the PPV tool can compare the effectiveness of various design features across different health conditions. For example, it can evaluate how curb ramp designs affect individuals with mobility issues differently compared to those with visual impairments. Policymakers can use the PPV tool to make data-driven decisions for targeted improvements in urban design. The tool’s insights into problematic areas can guide the allocation of resources where they are most needed and inform UD solutions to create more inclusive urban environments. The ultimate goal of UD is to enhance the user experience for everyone, regardless of their abilities. By addressing the specific barriers identified through the PPV tool, environments can be made more navigable, safer, and more comfortable for all users.

While this study provides valuable insights into the relationship between environmental barriers, health conditions, and routine activity performance through the development of the PPV score, there are several limitations to consider. One significant limitation is the generalization of the PPV based on the survey data, which has a relatively small sample size of 488 individuals. The skewed distribution of demographic characteristics, such as the disproportionate representation of males and adults younger than 29, could affect the data’s usability in the MNL approach. Also, the high percentage of missing income data poses challenges to the reliability of the analysis results. Moreover, the survey data primarily represent developed countries like the U.S., Canada, and Australia, limiting its generalizability to other developing nations with different built environments. Furthermore, the survey was conducted in 2008 and is therefore outdated. Significant societal changes, such as the COVID-19 pandemic and advancements in technology like driverless cars, may have influenced how people interact with the built environment since then. A new survey needs to be conducted to update this measure. The current internet-based PAS may include sample frame errors, so future surveys should consider multiple types of survey methods in the design for a new PPV development. However, the PAS data remain a valuable resource for understanding contemporary usability needs and evaluating interventions. Many challenges and barriers in the built environment persist over time, and the demographic information and health condition categorization in the dataset ensure its continued relevance by representing a diverse range of individuals with varying needs. Considering several limitations of this study, future studies should address these limitations by employing more robust methodologies, including larger and more diverse sample sizes and incorporating recent developments in society and technology. Finally, the current PPV tool focuses on activity and personal health components, excluding environmental factors. Future research should aim to integrate all three components into a comprehensive model to further enhance the effectiveness of UD solutions.

## 5. Conclusions

In conclusion, this study has contributed to our understanding of the complex interplay between environmental barriers, health conditions, and routine activity performance through the development of the PPV score. The PPV serves as a valuable usability measurement tool for individuals with diverse functional limitations, helping to predict problematic activities and identify areas for improvement in environmental design. By guiding the development of more inclusive urban spaces, the PPV analysis promotes equitable access for individuals with varying needs, ultimately enhancing user experiences and quality of life. Moving forward, it is imperative to address the study’s limitations and further refine the PPV tool to better inform UD solutions and improve the accessibility of built environments for all individuals.

## Figures and Tables

**Figure 1 ijerph-21-00764-f001:**
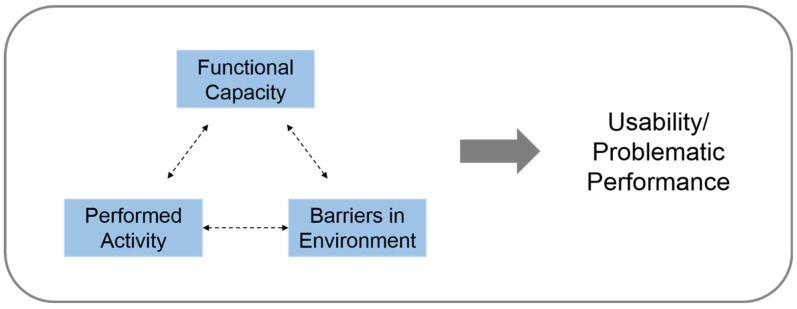
A conceptual framework for the usability measurement of user performance.

**Figure 2 ijerph-21-00764-f002:**
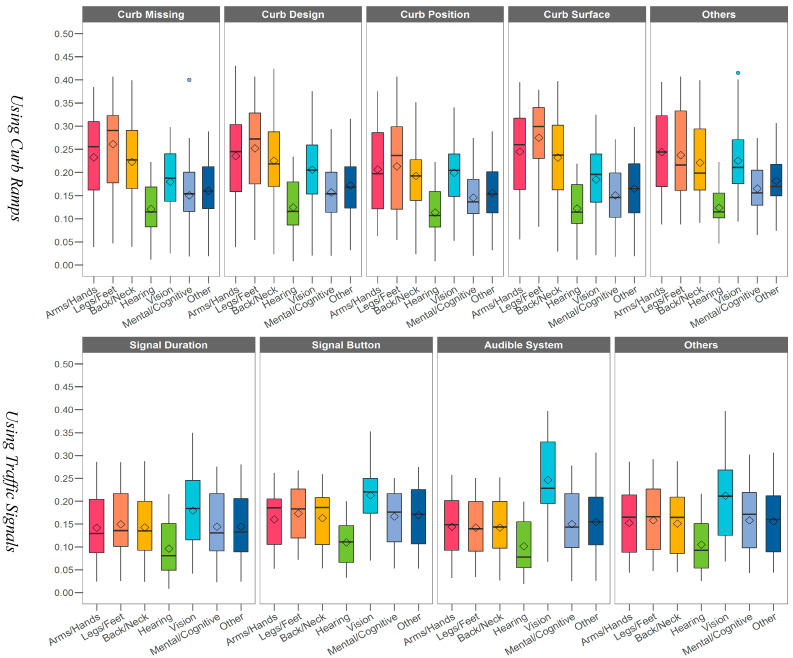
Boxplot of PPV scores for different health conditions and environmental barriers (from top, Using Curb Ramps; Using Traffic Signals; and Using Crosswalks).

**Table 1 ijerph-21-00764-t001:** Variables and basic descriptive statistics for MNL.

Variables	Definition	Mean (M-CURB/ M-SIGNAL/ M-CWALK)	Std Dev (M-CURB/ M-SIGNAL/ M-CWALK)	No. Obs (M-CURB/ M-SIGNAL/ M-CWALK)	Notes
Dependent Variable					
CURB	Categorical variable measuring a problem performing activities using curb ramps, 1 = Always, 2 = Sometimes, 3 = Never	2.54	0.59	478	M-CURB
SIGNAL	Categorical variable measuring a problem performing activities using pedestrian traffic signals, 1 = Always, 2 = Sometimes, 3 = Never	2.48	0.62	482	M-SIGNAL
CROSSWALK	Categorical variable measuring a problem performing activities using street crosswalks, 1 = Always, 2 = Sometimes, 3 = Never	2.58	0.58	478	M-CWALK
Independent Variables					
SEX	Respondent’s sex, 0 = Female, 1 = Male	0.33/0.33/0.33	0.47/0.47/0.47	478/482/478	All Models
MEDIAN_AGE	The median age of respondents from 14 categories (years)	45.15/45.29/45.13	15.75/15.78/15.72	478/482/478	All Models
AGE_SQ	Square of respondent’s median age (years)	2286/2300/2283	1438/1444/1433	478/482/478	All Models
INCOME_MD	Categorical variable describing annual household income level, 1 = if income is between USD 25,000 and USD 74,999; otherwise = 0	0.34/0.34/0.34	0.47/0.47/0.47	478/482/478	All Models
INCOME_HG	Categorical variable describing annual household income level, 1 = if income is USD 75,000 or more; otherwise = 0	0.28/0.28/0.28	0.45/0.45/0.45	478/482/478	All Models
INCOME_DN	Categorical variable describing annual household income level not reported, 1 = if prefer not to answer, don’t know, or not answered; otherwise = 0	0.13/0.13/0.13	0.33/0.33/0.33	478/482/478	All Models
H_TYPE	Dichotomous variable describing the type of housing, 1 = if single-family house; otherwise (multi-family house, apartment, condo, townhouse, mobile home, or missing value) = 0	0.58/0.57/0.58	0.49/0.49/0.49	478/482/478	All Models
ARM_AL	Categorical variable measuring how often mobility of arms/hands conditions affect the ability to perform routine activities, 1 = Always; otherwise = 0	0.17/0.16/0.16	0.37/0.37/0.37	478/482/478	Model 1
ARM_ST	Categorical variable measuring how often mobility of arms/hands conditions affect ability to perform routine activities, 1 = Sometimes; otherwise = 0	0.27/0.27/0.27	0.44/0.44/0.44	478/482/478	Model 1
LEG_AL	Categorical variable measuring how often mobility of legs/feet conditions affect ability to perform routine activities, 1 = Always; otherwise = 0	0.32/0.32/0.32	0.47/0.47/0.47	478/482/478	Model 2
LEG_ST	Categorical variable measuring how often mobility of legs/feet conditions affect ability to perform routine activities, 1 = Sometimes; otherwise = 0	0.2/0.2/0.2	0.4/0.4/0.4	478/482/478	Model 2
BACK_AL	Categorical variable measuring how often mobility of back/neck conditions affect ability to perform routine activities, 1 = Always; otherwise = 0	0.13/0.13/0.13	0.34/0.34/0.34	478/482/478	Model 3
BACK_ST	Categorical variable measuring how often mobility of back/neck conditions affect ability to perform routine activities, 1 = Sometimes; otherwise = 0	0.33/0.33/0.33	0.47/0.47/0.47	478/482/478	Model 3
HEAR_AL	Categorical variable measuring how often hearing conditions affect ability to perform routine activities, 1 = Always; otherwise = 0	0.12/0.12/0.12	0.33/0.33/0.32	478/482/478	Model 4
HEAR_ ST	Categorical variable measuring how often hearing conditions affect ability to perform routine activities, 1 = Sometimes; otherwise = 0	0.23/0.24/0.23	0.42/0.43/0.42	478/482/478	Model 4
SIGHT_AL	Categorical variable measuring how often sight conditions affect ability to perform routine activities, 1 = Always; otherwise = 0	0.14/0.15/0.15	0.35/0.35/0.35	478/482/478	Model 5
SIGHT_ST	Categorical variable measuring how often sight conditions affect ability to perform routine activities, 1 = Sometimes; otherwise = 0	0.25/0.25/0.25	0.44/0.43/0.43	478/482/478	Model 5
MENTAL_AL	Categorical variable measuring how often mental and/or cognitive conditions affect ability to perform routine activities, 1 = Always; otherwise = 0	0.05/0.05/0.04	0.21/0.21/0.21	478/482/478	Model 6
MENTAL_ ST	Categorical variable measuring how often mental and/or cognitive conditions affect ability to perform routine activities, 1 = Sometimes; otherwise = 0	0.13/0.13/0.13	0.33/0.34/0.33	478/482/478	Model 6
OTHER_AL	Categorical variable measuring how often other health conditions affect ability to perform routine activities, 1 = Always; otherwise = 0	0.11/0.11/0.11	0.31/0.31/0.31	478/482/478	Model 7
OTHER_ ST	Categorical variable measuring how often other health conditions affect ability to perform routine activities, 1 = Sometimes; otherwise = 0	0.22/0.22/0.22	0.41/0.42/0.41	478/482/478	Model 7

**Table 2 ijerph-21-00764-t002:** Decomposed structures of individual probabilities of a specific PAS activity for one health condition.

		Probabilities of Problematic Performance		
		Always	Sometimes	Never
Health Condition (Individual Selection)		P_h_ (Always)	P_h_ (Sometimes)	P_h_ (Never)
Health Condition	Always	P_11_	P_12_	P_13_
	Sometimes	P_21_	P_22_	P_23_
	Never	P_31_	P_32_	P_33_

**Table 3 ijerph-21-00764-t003:** The parameter estimates of the model of Using Ramp Curbs (M-CURB).

		Model 1	Model 2	Model 3	Model 4	Model 5	Model 6	Model 7
Level		Est. Coef.	Est. Coef.	Est. Coef.	Est. Coef.	Est. Coef.	Est. Coef.	Est. Coef.
Intercept	1	−6.214 ***	−7.419 **	−6.709 **	−6.087 **	−6.522 **	−6.492 **	−5.605 **
	2	−6.076 ***	−6.204 ***	−6.114 ***	−6.183 ***	−6.505 ***	−6.415 ***	−6.383 ***
median_age	1	0.126	0.110	0.147	0.172	0.177	0.177	0.129
	2	0.227 ***	0.231 ***	0.233 ***	0.257 ***	0.262 ***	0.268 ***	0.262 ***
age_SQ	1	−0.001	−0.001	−0.001	−0.002	−0.002	−0.002	−0.001
	2	−0.002 ***	−0.002 ***	−0.002 ***	−0.002 ***	−0.003 ***	−0.003 ***	−0.003 ***
Sex	1	−1.773 *	−1.872 *	−1.634 *	−1.697 *	−1.568 *	−1.588 *	−1.696 *
	2	0.065	−0.101	0.069	−0.040	0.100	0.067	0.083
income_md	1	−0.757	−0.877	−0.573	−0.835	−0.910	−0.923	−0.595
	2	−0.127	−0.167	−0.082	−0.151	−0.240	−0.226	−0.188
income_hg	1	−2.786 *	−2.751 *	−2.655 *	−3.238 **	−3.154 **	−2.989 **	−2.850 *
	2	−0.429	−0.395	−0.475	−0.725 *	−0.685 *	−0.755 *	−0.669 *
income_dn	1	0.091	0.249	0.069	−0.350	−0.395	−0.269	−0.079
	2	−0.626	−0.576	−0.715	−0.794 *	−0.906 *	−0.940 *	−0.878 *
h_type	1	0.867	1.016	1.041	0.994	1.095 *	0.894	1.102 *
	2	−0.142	−0.126	−0.097	−0.100	0.002	−0.075	−0.085
HC_al	1	2.913 ***	4.501 ***	2.846 ***	−0.509	1.486 *	1.911 **	1.601 **
	2	1.480 ***	1.923 ***	1.371 ***	−1.015 **	1.160 ***	−0.007	0.202
HC_st	1	1.111	2.019	0.752	−0.051	0.338	0.408	0.342
	2	0.893 ***	0.190	0.541 *	−0.188	0.005	−0.218	0.078
No. Obs		478	478	478	478	478	478	478
−2LL		658.114	611.532	670.647	695.371	686.464	695.380	697.279

Notes: 1. * *p* < 0.05, ** *p* < 0.01, *** *p* < 0.001; 2. Level 1: Always vs. Never (ref), Level 2: Sometimes vs. Never (ref) in encountering problematic performance using curb ramps; 3. −2LL = −2 Log Likelihood; Est. Coef. = Estimated Coefficients.

**Table 4 ijerph-21-00764-t004:** The parameter estimates of the model of Using Traffic Signals (M-SIGNAL).

		Model 1	Model 2	Model 3	Model 4	Model 5	Model 6	Model 7
Level		Est. Coef.	Est. Coef.	Est. Coef.	Est. Coef.	Est. Coef.	Est. Coef.	Est. Coef.
Intercept	1	−3.981 *	−3.889 *	−4.076	−3.992 *	−4.339 *	−4.183 *	−3.974 *
	2	−2.859 ***	−2.747 **	−2.818 ***	−2.586 **	−2.869 ***	−2.981 ***	−2.981 ***
median_age	1	0.066	0.063	0.071	0.073	0.053	0.082	0.064
	2	0.098 *	0.091 *	0.095 *	0.089 *	0.091 *	0.106 **	0.104 **
age_SQ	1	−0.0004	−0.0003	−0.0004	−0.0004	−0.0003	−0.001	−0.0004
	2	−0.001	−0.001	−0.001	−0.001	−0.001	−0.001 *	−0.001 *
Sex	1	−0.823	−0.830	−0.770	−0.838	−0.715	−0.710	−0.713
	2	0.100	0.059	0.083	−0.124	0.093	0.076	0.087
income_md	1	−0.120	−0.142	−0.094	−0.147	−0.349	−0.187	−0.107
	2	0.261	0.268	0.264	0.345	0.235	0.280	0.260
income_hg	1	−0.699	−0.683	−0.682	−0.902	−0.751	−0.812	−0.674
	2	−0.470	−0.452	−0.481	−0.571	−0.483	−0.467	−0.457
income_dn	1	0.099	0.084	0.090	0.008	0.005	−0.015	0.076
	2	−0.418	−0.383	−0.433	−0.351	−0.451	−0.427	−0.453
h_type	1	−0.138	−0.050	−0.028	−0.017	0.231	−0.042	−0.010
	2	−0.126	−0.145	−0.134	−0.129	−0.036	−0.157	−0.143
HC_al	1	0.926	0.609	0.697	−0.750	2.971 ***	0.303	0.657
	2	0.106	0.378	0.147	−1.545 ***	1.534 ***	0.111	0.127
HC_st	1	0.010	−0.831	−0.179	0.183	0.319	0.170	0.445
	2	0.305	0.346	0.350	0.081	0.371	0.259	0.454
No. Obs		482	482	482	482	482	482	482
−2LL		805.663	802.823	806.093	791.465	763.376	810.976	807.126

Notes: 1. * *p* < 0.05, ** *p* < 0.01, *** *p* < 0.001; 2. Level 1: Always vs. Never (ref), Level 2: Sometimes vs. Never (ref) in encountering problematic performance using traffic signals; 3. −2LL= −2 Log Likelihood; Est. Coef. = Estimated Coefficients.

**Table 5 ijerph-21-00764-t005:** The parameter estimates of the model of Using Crosswalks (M-CWALK).

		Model 1	Model 2	Model 3	Model 4	Model 5	Model 6	Model 7
Level		Est. Coef.	Est. Coef.	Est. Coef.	Est. Coef.	Est. Coef.	Est. Coef.	Est. Coef.
Intercept	1	−5.842 **	−5.749 **	−6.031 **	−6.141 **	−6.513 **	−6.193 **	−6.007 **
	2	−4.735 ***	−4.681 ***	−4.728 ***	−4.855 ***	−5.071 ***	−5.044 ***	−5.092 ***
median_age	1	0.130	0.120	0.148	0.154	0.152	0.149	0.134
	2	0.173 ***	0.166 ***	0.170 ***	0.194 ***	0.195 ***	0.203 ***	0.203 ***
age_SQ	1	−0.001	−0.001	−0.001	−0.001	−0.001	−0.001	−0.001
	2	−0.002 **	−0.002 **	−0.002 **	−0.002 ***	−0.002 ***	−0.002 ***	−0.002 ***
Sex	1	−0.621	−0.567	−0.488	−0.482	−0.437	−0.399	−0.404
	2	−0.275	−0.357	−0.269	−0.368	−0.244	−0.272	−0.247
income_md	1	−0.572	−0.653	−0.632	−0.727	−0.842	−0.653	−0.634
	2	−0.120	−0.138	−0.051	−0.146	−0.232	−0.180	−0.201
income_hg	1	−2.112 *	−2.117 *	−2.223 *	−2.367 **	−2.304 **	−2.128 *	−2.080 *
	2	−1.096 ***	−1.082 **	−1.053 **	−1.311 ***	−1.286 ***	−1.285 ***	−1.268 ***
income_dn	1	0.041	0.048	−0.117	−0.234	−0.159	−0.066	−0.082
	2	−0.497	−0.465	−0.510	−0.609	−0.686	−0.689	−0.707
h_type	1	0.124	0.251	0.295	0.304	0.454	0.196	0.292
	2	−0.139	−0.122	−0.110	−0.087	0.011	−0.090	−0.096
HC_al	1	1.468 **	1.255 *	0.603	−0.037	2.139 ***	0.799	0.723
	2	0.985 **	1.242 ***	1.142 ***	−0.824 *	1.201 ***	−0.328	−0.036
HC_st	1	−0.120	−0.036	−0.208	0.356	0.034	0.624	0.920
	2	0.557 *	0.340	0.520 *	−0.068	0.093	0.015	0.242
No. Obs		478	478	478	478	478	478	478
−2LL		675.669	664.896	679.119	687.311	668.709	691.077	689.565

Notes: 1. * *p* < 0.05, ** *p* < 0.01, *** *p* < 0.001; 2. Level 1: Always vs. Never (ref), Level 2: Sometimes vs. Never (ref) in encountering problematic performance using crosswalks; 3. −2LL= −2 Log Likelihood; Est. Coef. = Estimated Coefficients.

**Table 6 ijerph-21-00764-t006:** Mean and median PPV scores by health conditions and activities.

			Health Condition						
			Arms	Legs	Back	Hearing	Vision	Mental	Other
Mean PPV Score	Activities	Curb	0.169	0.162	0.165	0.095	0.161	0.124	0.135
		Signal	0.133	0.139	0.133	0.086	0.171	0.137	0.136
		Crosswalk	0.126	0.123	0.132	0.082	0.131	0.092	0.103
Median PPV Score	Activities	Curb	0.162	0.151	0.169	0.097	0.173	0.129	0.142
		Signal	0.119	0.124	0.119	0.074	0.150	0.123	0.119
		Crosswalk	0.106	0.110	0.112	0.056	0.103	0.060	0.075

## Data Availability

The data presented in this study are available on request from the corresponding author.

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
