# Peer review of "Developing Problematic Performance Value Scores: Binding Routine Activity Performance, Environmental Barriers, and Health Conditions"

_ijerph, 2024, doi:10.3390/ijerph21060764_

Round 1

Reviewer 1 Report

Comments and Suggestions for Authors

Binding Routine Activity Performance, Environmental Barriers, and Health Conditions: Multinomial Logit Application To Generalizing Problematic Performance Value Scores

The manuscript explored the relationships among environmental barriers, health conditions, and routine activity performance, and developed an index to evaluate users’ universal design (UD) performance in a functional capacity, it is an interesting topic. Upon reviewing the manuscript, some issues have come to my attention as the below:

Abstract

The abbreviation "UD" should be accompanied by its full name upon first mention. Additionally, the review lacks clarification on the data source, the characteristics of the study population, and the regional context of the study area.

1.     Introduction

In the entire introduction section, the most recent literature cited is from the demographic statistics of the United States in 2018 and 2019. Even though there is a clear explanation of the theoretical gap, considering that the literature on UD and environmental barriers in the last three years has also been updated, it is recommended to incorporate some recent studies from the last three years to enhance the credibility of addressing research gaps and emphasize the significance of this study in the context of recent literature.

Explain clearly what does it mean by UD in this study.

2. Methods

2.1. Data Collection: While the manuscript indicates that an anonymized and non-personally identifiable information dataset was acquired from the Problematic Activities Survey (PAS) conducted in 2008, it lacks a comprehensive background introduction to this survey (such as survey objective, survey participants, data collection process et al.). Additionally, the survey was conducted in 2008, and considering that it is now 2024, while the limitation section acknowledges this disadvantage, it is still advisable to provide justification for the applicability of this study's data to contemporary research.

2.2. Environmental Barriers

The Problematic Activities Survey (PAS) conducted in 2008 focused on three environmental settings: public streets, public buildings, and residential environments. However, this study specifically concentrated on intersection activities on public streets, including Using Curb Ramps (Curb), Using Traffic Signals (Signal), and Using Crosswalks (Crosswalks). While the manuscript lacks clear support for the selection of public streets as the focus area, it is important to clarify the scientific rationale behind this choice. I recommend revising the introduction or relevant sections to provide robust justification for the selection of the study area, enhancing the scientific rigour of the research.

2.3. Participant Demographic Characteristics:

The introduction section has a paragraph which leans towards portraying the study population as elderly (refer to the population background of the United States). However, the methods section indicates that the study covers all age groups and includes participants not only from the United States but also from Canada and Australia. This discrepancy could confuse the clarity regarding the specific study population and its background. It is advisable to clarify the logical connection between these two descriptions and clearly specify the study population and background of this manuscript.

Discussion / Conclusion

The Results section provides a detailed presentation of the data analysis, and the Discussion and Conclusion sections elaborate extensively on the application or value of the PPV proposed by the study. However, some of the research findings seem to merely reiterate common knowledge or everyday understanding, which significantly diminishes the value of the study results. For instance: The distributions of a wide range of PPVs for arms/hands, legs/feet, back/neck, and vision for activity Curb further indicated more significant usability problems these populations with more severe disabilities would encounter. Individuals having sight difficulty seemed to have greater usability problems when encountering issues with the audible signal system. Moreover, if the PPV score is used solely to assess issues that are already commonly understood, the theoretical and practical value of the PPV is not highlighted. It is advisable to reiterate the novelty of the study results emphasized in this manuscript and use these findings to illustrate the value of PPV, highlighting the theoretical and practical value that this study can bring in achieving its research objectives.

The discussion section should focus more on the interpretation of research results, discussion of research limitations, and future prospects. In the conclusions section, it is advisable to streamline the summary of the entire research process, research findings, and their implications. A concise conclusion that effectively closes the loop on the entire study will make it clearer and more comfortable for readers to follow.

Author Response

Dear Reviewer,

My co-author and I now provide a revised manuscript that addresses a set of comments raised. We sincerely appreciate you for the patience and valuable comments, which significantly improved the manuscript, and we hope this revision can satisfy your comments. Please find the detailed responses in the attachment and the corresponding revisions/corrections in track changes in the re-submitted files. Thank you. 

Reviewer 2 Report

Comments and Suggestions for Authors

Comments on the Quality of English Language

Author Response

(The authors gave the same response as above.)

Reviewer 3 Report

Comments and Suggestions for Authors

This interesting research explores the relation between environmental barriers and physical activity rate among users with certain types of health conditions. Accordingly, their functional mobility is evaluated considering the problematic performance value scores.    

However, I have ethical concerns regarding this manuscript since same authors have published the same content in Jan 2024:  

https://papers.ssrn.com/sol3/papers.cfm?abstract_id=4705036

Also, please address the following points please: 

- Title is to be presented in fewer words. 

- [UD] is to be written in its full format [Universal Design] when used in the Abstract for the first time in the manuscript.

- How was the sampling group calculated to reach the sufficient respondents?

- The research is based on a very limited references; while only one of them belongs to the last five years.

(except Ref 27and Ref 12, all others are published before 2018). 

- Discussion is very limited and it is not even developed based on the (few) presented reviewed literature. 

- How were physical disabilities measured? Was it through presenting any medical certificate? Were any consideration about 

temporal and permanent disabilities? (This is due to the different attitudes and behaviors and their adaptability skills.)

Regards

Author Response

My co-author and I now provide a revised manuscript that addresses a set of comments raised. We sincerely appreciate you for the patience and valuable comments, which significantly improved the manuscript, and we hope this revision can satisfy your comments. Please find the detailed responses in the attachment and the corresponding revisions/corrections in track changes in the re-submitted files. 

Round 2

Reviewer 1 Report

Comments and Suggestions for Authors

Comments to authors,

I am happy with most of the changes made, as the revised manuscript has shown significant improvement. However, providing a more robust justification for the applicability of data from the 2008 survey to contemporary research is essential. The justifications provided in the Cover Letter are more convincing compared to those provided in the revised manuscript. Therefore, incorporating such justifications into the manuscript would enhance its methodological rigor and strengthen the validity of its findings.

Author Response

My co-author and I now provide a second revision of this manuscript, which addresses a set of comments raised. We sincerely appreciate you for additional comment and hope this revision can satisfy your comment. The corresponding revisions/corrections are found in track changes in the re-submitted files.

Responses to Reviewer 1

I am happy with most of the changes made, as the revised manuscript has shown significant improvement. However, providing a more robust justification for the applicability of data from the 2008 survey to contemporary research is essential. The justifications provided in the Cover Letter are more convincing compared to those provided in the revised manuscript. Therefore, incorporating such justifications into the manuscript would enhance its methodological rigor and strengthen the validity of its findings.

--> We appreciate the review comments that improved the manuscript and thanks for the suggestion. We modified and updated the data justification, detailing the usefulness of the survey data. Connecting it to the methodological rigor and implication of this study, we tried to strengthen the validity of this study; this is highlighted the part in the last paragraph of Introduction.

Reviewer 2 Report

Comments and Suggestions for Authors

Review Comments on "Developing Problematic Performance Value Scores:Binding Routine Activity Performance, Environmental Barriers, and

Health Conditions"

The article innovatively uses the PPV framework to quantitatively analyze the relationships between personal functional capacities, environmental barriers, and performed activities to aid in the universal design (UD) of public spaces. While the overall logic of the article is clear, some sections have issues with expression and logic. The specific comments are as follows:

1.       The introduction section is relatively clear and explains the significance of the study. However, it lacks a literature review on the interactions between individual abilities, environmental barriers, and performed activities. It is recommended to include a review of existing research to support the study.

2.       The conclusion of the introduction could be enhanced by combining the study's significance and research methods to highlight the article's innovations. The description of the innovative aspects of the article is insufficient.

3.       In the data collection section, it is recommended to clearly specify the area covered by the data collection and indicate the study area.

4.       In the discussion section, lines 376-388 describe the advantages of the research methods used in the article. It is recommended to move this content to the introduction section, as the earlier part of the text lacks relevant details.

5.       The discussion section lacks sufficient analysis of the results. It is recommended to enhance the content on the causes and improvement strategies, and to increase the explanation of how the study's results contribute to the usability and practicality of Universal Design (UD).

Author Response

Dear Reviewer 2:

Thanks for the comments and please see our replies attached. 

Sincerely.

Reviewer 3 Report

Comments and Suggestions for Authors

Thanks for addressing the comments and revisions to the manuscript. 

Author Response

Dear Reviewer 3:

Many thanks for your review. We could improve our paper with your comments. Appreciate it again.